# SCALABLE SUBSET SAMPLING WITH NEURAL CONDITIONAL POISSON NETWORKS

**Adeel Pervez**
QUVA Lab,
Informatics Institute
University of Amsterdam
a.a.pervez@uva.nl

**Phillip Lippe**
QUVA Lab,
Informatics Institute
University of Amsterdam
p.lippe@uva.nl

**Efstratios Gavves**
QUVA Lab,
Informatics Institute
University of Amsterdam
e.gavves@uva.nl

## ABSTRACT

A number of problems in learning can be formulated in terms of the basic primitive of sampling $k$ elements out of a universe of $n$ elements. This subset sampling operation cannot directly be included in differentiable models, and approximations are essential. Current approaches take an *order sampling* approach to sampling subsets and depend on differentiable approximations of the Top-$k$ operator for selecting the largest $k$ elements from a set. We present a simple alternative method for sampling subsets based on *conditional Poisson sampling*. Unlike order sampling approaches, the complexity of the proposed method is independent of the subset size, which makes the method scalable to large subset sizes. We adapt the procedure to make it efficient and amenable to discrete gradient approximations for use in differentiable models. Furthermore, the method allows the subset size parameter $k$ to be differentiable. We validate our approach extensively, on image and text model explanation, image subsampling and stochastic $k$-nearest neighbor tasks outperforming existing methods in accuracy, efficiency and scalability.

## 1 INTRODUCTION

The fundamental combinatorial operation of selecting subsets of elements from a given universe is ever increasingly being incorporated in differentiable neural models due to its range of applicability. Example applications include model explanations (Chen et al., 2018), sequence modeling (Kool et al., 2019), point cloud modeling (Yang et al., 2019), and nearest neighbor networks (Grover et al., 2018).

Current neural network approaches for sampling subsets generally fall in the class of *order sampling* methods. In the order sampling scheme, each element in the universe is assigned an independent *ranking* random variable. To obtain a subset sample of size $k$, the largest (or smallest) $k$ elements are chosen. Thereby, the ranking variable distribution induces a probability distribution over the possible subsets. However, the operation of choosing the largest $k$ elements (Top-$k$) is naturally not differentiable, since it is a discrete operation. This means that the Top-$k$ procedure cannot be directly used in gradient learning models. This has led to a number of proposals of relaxed and differentiable versions of the Top-$k$ operator (Goyal et al., 2018; Pietruszka et al., 2021; Plötz & Roth, 2018). Building on Top-$k$ approaches several methods of sampling subsets as $k$-hot vectors have appeared in the literature (Paulus et al., 2020; Xie & Ermon, 2019).

In this paper, we explore *Poisson sampling* (Tillé, 2006) and *conditional Poisson sampling* (Hájek & Dupač, 1981) as an alternative to order sampling for subsets. With Poisson sampling, each element in the set is *independently* drawn to be selected for the subset or not. As these independent trials cannot guarantee a fixed size for subsets, with conditional Poisson sampling, the Poisson sampling procedure is conditioned to return subsets of exactly $k$ elements. In practice, the conditioning amounts to repeating the Poisson sampling procedure until a subset of size $k$ is obtained.

The general (conditional) Poisson sampling approach has a number of features which make it an attractive alternative to Top-$k$-based order sampling methods. Firstly, the sampling is done independently in Poisson sampling methods, which makes the procedure very efficient for sampling subsets with large values of $k$. By contrast, current Top-$k$ methods (Goyal et al., 2018; Plötz & Roth, 2018) often have an inner loop depending on $k$, which makes them expensive for sampling large

subsets in terms of both time and memory. Furthermore, computations in modern neural network models are vectorized. This makes sampling different subset size $k$ for different elements in a batch difficult for current Top-$k$ procedures, since the number of sampling iterations to obtain the Top-$k$ elements depend on $k$. With Poisson sampling, it is trivial to sample different subset sizes for batched inputs, making it ideally suited to vectorized computation. Finally, with Top-$k$, the subset size $k$ itself is not differentiable. With Poisson sampling, $k$ appears as a scaling parameter for the probabilities of the individual elements in the universe. Therefore, the subset size parameter $k$ can easily be incorporated in differentiable computations when having a differentiable sampling procedure.

Despite the aforementioned advantages, there are two difficulties with Poisson sampling. The first is that vanilla Poisson sampling can lead to large variance in the sampled subset size. This can be resolved with conditional Poisson sampling to obtain exact samples, but only at the cost of high computational complexity. The second (and main) difficulty is that neither Poisson sampling nor its conditional variant are differentiable and cannot be directly included in differentiable models.

In this paper, and in the context of differentiable subset sampling with neural networks, we propose *neural conditional Poisson subset sampling*. We note that often we do not need subsets of $k$ elements exactly, as conditional Poisson sampling would have us do, and instead sampling $k$-sized subsets in expectation is enough. With neural conditional Poisson subset sampling, we relax the constraint of sampling exactly $k$ elements, thus allowing to trade off accuracy in the subset size for efficiency of sampling large subsets. Compared to Top-$k$ approaches for sampling subsets (Xie & Ermon, 2019), neural conditional Poisson subset sampling allows for efficient sampling of large subsets, easy choice of per-instance subset sizes, and differentiable subset sizes for a small loss in subset size accuracy, when an exact number of elements in the sampled subsets is not a necessity. Secondly, we adapt the sampling procedure so that gradient approximations for discrete variables are applicable. The resulting method is scalable and can be used to sample large subsets even from image-size domains in full resolution – a task that is to date infeasible for current subset sampling methods.

## 2 PRELIMINARIES

Let $U = \{1, 2 \ldots, n\}$ denote a universe consisting of $n$ elements. Each element $i \in U$ is assigned a "size" $p_i \in (0, 1)$. We assume the sizes here to be normalized to the unit interval. Let $x$ denote a subset of the elements in $U$ represented as an indicator vector of size $n$, $x = (x_1, \ldots, x_n)$, where $x_i \in \{0, 1\}$ and $x_i = 1$ if the $i$th element is included in the subset. The *sample size* is the number of elements, i.e., $\sum x_i$, in the chosen subset. In this paper, we are concerned with sampling subsets of given size $k$ from a universe of $n$ elements.

A *sampling design* (Tillé, 2006), $S$, is a way to assign a probability to each subset of universe $U$, i.e., $S : \mathcal{P}(U) \rightarrow [0, 1]$, where $\mathcal{P}(U)$ is the power set. Intuitively, a sampling procedure induces a sampling design by assigning each subset with the probability with which the subset is chosen. Conversely, there could be a number of sampling procedures that correspond to the same sampling design. Occasionally, a sampling procedure is also referred to as a sampling design.

**Inclusion Probability.** Important parameters of a sampling design are the inclusion probabilities. The *first order inclusion probability* of an element $i$ is the marginal probability, over the space of samples, that $i$ is included in the sample. If $I_i$ is an indicator variable where $I_i = 1$ when $i$ is included, the $i$-th inclusion probability is $\pi_i := \mathbb{E}[I_i]$.

### 2.1 POISSON SAMPLING

Poisson sampling (Tillé, 2006) is a *probability-proportional-to-size* sampling design for sampling without replacement. This means that each element $i$ is included in the sample with probability proportional to its size $p_i$, where we assume that $\sum_i p_i = 1$. With Poisson sampling, the sample size is a random variable with expected size $k$. Given independent uniform random variables $u_i \sim \mathcal{U}(0, 1)$, an element $i$ is included in the sample if $u_i \leq kp_i$ (see Algo-

---

**Algorithm 1** Poisson Sampling

---

**Require:** Input $p \in (0, 1)^n$, $\sum_i p_i = 1$; $k$ integer;
**Require:** Output $S$
1: **for** $i = 1, ..., n$ **do**
2:     Sample $u_i \sim \mathcal{U}(0, 1)$
3:     **if** $u_i \leq kp_i$ **then**
4:         $S \leftarrow S \cup \{i\}$
5:     **end if**
6: **end for**

---

rithm 1). When $kp_i > 1$ for some $i$, the corresponding elements are always added to the output. The procedure is then repeated for the remaining elements after appropriate normalization (Tillé, 2006). When using Poisson sampling, the resulting sample can have large variance in sample size due to independent sampling of elements.

## 2.2 CONDITIONAL POISSON SAMPLING

Conditional Poisson sampling (Hájek & Dupač, 1981) is a sampling design over samples with exactly $k$ elements. The design can be implemented simply by repeating the Poisson sampling procedure until a sample with exactly $k$ elements is obtained. That is, discarding the entire sampled subset if it does not contain exactly $k$ element and repeating from scratch. Since the sampling has to be repeated, it may require many trials before a size $k$ sample is obtained. Hájek & Dupač (1981) showed that conditional Poisson sampling is a maximum entropy design, subject to *required* inclusion probabilities $p_i$ (when $\sum p_i = k$) and sample size $k$.

---

**Algorithm 2** Conditional Poisson Sampling

**Require:** Input $p \in (0,1)^n$, $\sum_i p_i = 1$; $k$ integer;
**Require:** Output $S$
1: $S \leftarrow \{\}$
2: **while** $|S| \neq k$ **do**
3:     $S \leftarrow \{\}$
4:     **for** $i = 1, ..., n$ **do**
5:         Sample $u_i \sim \mathcal{U}(0,1)$
6:         **if** $u_i \leq kp_i$ **then**
7:             $S \leftarrow S \cup \{i\}$
8:         **end if**
9:     **end for**
10: **end while**

---

One potential difficulty with conditional Poisson sampling is that the inclusion probabilities $\pi_i$, which we would approach if we repeated the sampling numerous times, are *approximations* of the desired probabilities $p_i$, when $p_1, ..., p_n$ are also used as sampling probabilities. This is because with standard Poisson sampling with the probabilities $p_i$, not all samples result in a valid subset of size $k$.

It is possible to improve the approximation by correcting the given $p_i$ to obtain sampling probabilities $p'_i$. Using the corrected $p'_i$s for sampling leads to inclusion probabilities, $\pi_i$, that are closer to the desired probabilities $p_i$ (Lundquist, 2009).

# 3 NEURAL CONDITIONAL POISSON SUBSET SAMPLING

We first give an overview of the proposed sampling procedure, and we detail it next.

## 3.1 OVERVIEW

Conditional Poisson sampling only accepts samples of exactly $k$ elements, discarding the sample otherwise. This may require numerous sampling iterations, thus becoming an efficiency bottleneck. On the other hand, vanilla Poisson sampling can be viewed as repeating the same process only once, thus yielding high variance in the subset size. Furthermore, neither Poisson sampling nor the conditional extension is differentiable relative to the parameters $p_i$. Oftentimes, it is not strictly necessary to sample exactly size $k$ subsets (unlike conditional Poisson sampling), for as long as the subsets that we sample are size $k$ in expectation and with low variance (unlike Poisson sampling). This, for example, is the case when we need to explain images in terms of large subsets of pixels.

Leveraging this observation, we propose neural conditional Poisson subset sampling (NCPSS), which is intermediate between Poisson sampling and conditional Poisson sampling. Similar to conditional Poisson sampling, our sampling procedure performs multiple passes. Differently, however, for every new pass, and depending on whether our sample has more or less than $k$ elements, we do not throw away the previously sampled subset: instead, we add or remove elements from it. Crucially, we repeat our sampling procedure only for a predetermined number of passes. That is, unlike conditional Poisson sampling, we do not need to sample exactly $k$ elements to terminate. We show that this adaptation reduces variance in the subset size while yielding significant gains in sampling efficiency.

## 3.2 PROPOSED SAMPLING PROCEDURE

In the context of neural networks, we want to make our sampling procedure amenable to training by discrete variable gradient approximations. We, therefore, first adapt the Poisson sampling procedure

so that the scaled probabilities $kp_i$ remain bounded in $(0, 1)$. That is, if we have sampling probabilities $p_i$ and their sum $s := \sum_i p_i$, we compute the normalized probabilities $p_i/s$. The reason is that we then only require for our new sampling procedure to sample from Bernoulli distributions, and we can rely on popular gradient estimations for discrete operations, like Straight-Through (Bengio et al., 2013) or Gumbel-Sigmoid relaxations (Maddison et al., 2017).

**Reversing samples.** To sample a subset of size $k$ with Poisson sampling, we multiply the normalized probabilities by $k$. Unfortunately, this results in probabilities $p_i k/s$ that can be greater than 1 if $k/s > 1$. Specifically, elements $i$ with $p_i k/s > 1$ will certainly be included in the sample, which, however, is no longer a valid Bernoulli sample since the "probability" parameter is greater than 1.

We make, however, the observation that if $k/s > 1$, we could still sample from the complementary distribution with probabilities $1 - p_i$, since $\sum(1 - p_i) = n - s$ and $(n - k)/(n - s) \leq 1$ where $n$ is the total number of elements. This implies that if $k > s$, we can perform Poisson sampling with probabilities $(1 - p_i)$ to compute the individual sampling probabilities as $(1 - p_i)(n - k)/(n - s)$. Performing Bernoulli sampling with these new probabilities gives a sample with expected size $n - k$. In this case, rather than sampling the $k$ elements to include from our full set, we sample instead the $n - k$ elements (in expectation) to exclude. For the final sample, we simply flip (i.e., flip 1s and 0s) our obtained "exclusion" sample of expected size $k$, having used only Bernoulli samples.

**Reducing variance by iterating.** Our new sampling procedure is based on Poisson sampling, and thus will return samples of size $k$ in expectation. To reduce variance in sample size, we perform a (small) prescribed number of passes. Let $U = \{1, ..., n\}$ be our full set and $S_t$ the currently sampled subset at iteration $t$, During each new pass $t+1$, we either add or remove elements from $S_t$ to get closer to size $k$. Specifically, if our current sampled subset is smaller than it should ($|S_t| < k$), we make a new pass over the unselected elements, $U - S_t$, to sample new elements to add. Conversely, if our current sampled subset is larger than it should ($|S_t| > k$), we make a new pass over the currently selected elements $S_t$, and use the inverted probabilities, $1 - p_i$, to sample which elements to remove.

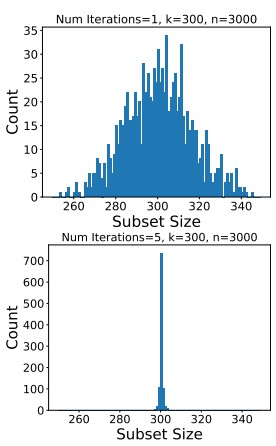

While this procedure is more efficient than conditional Poisson sampling by relaxing the constraint of obtaining exactly size-$k$ subsets, we can also show that it produces samples of lower variance compared to regular Poisson sampling. Specifically, under reasonable assumptions and for the simpler case of equal input probabilities, i.e., $p_1 = p_2 = ... = p_n$, the subset size variance decreases exponentially with the number of iterations. This implies that only a few iterations are required to obtain samples that are close to the target size $k$.

**Proposition 3.1.** *Let $S_i \in \{0, 1\}^n$ denote the subset and $q_i$ bounding the probability at step $i$ in the iterative Poisson sampling procedure. Then after $T$ iterations we have $Var(|S_T|) \leq Var(|S_1|) \prod_{i=2}^{T} q_i$, where $Var(|S_1|)$ is the Poisson sampling variance. Assuming that the probability bound is bounded away from 1, i.e., $q_i < 1 - \epsilon$ for $\epsilon > 0$, the procedure obtains an exponential decrease in variance with the number of steps $T$.*

Figure 1: Histograms showing subset size distribution with $t = 1$ iteration (top) and 5 iterations (bottom), when choosing $k = 300$ elements from $n = 3000$ elements. We observe a large reduction in variance with 5 iterations.

We provide the proof of this proposition in Appendix B, as well as an empirical verification of the exponential reduction in variance in Figure 1 with the details of the experiment given in Appendix E.

## 3.3 END-TO-END ALGORITHM

Next, we introduce Neural Conditional Possion Subset Sampling (NCPSS), a differentiable way of using the previously described sampling procedure in a gradient-based learning framework. Given a probability vector $p_i$ parameterized by a neural network and sum $s = \sum_i p_i$, for each $i$, we adjust the sampling probabilities to the subset size $s$. We refer to these probabilities as $q_i$ and compute them as:

$$q_i = \begin{cases} kp_i/s & k <= s \\ 1 - (n - k)(1 - p_i)/(n - s) & k > s \end{cases} \tag{1}$$

---

**Algorithm 3** Iterative Poisson Sampling

**Require:** Input $p \in (0,1)^n$; $k$ integer; $t$ iterations
 1: Initialize empty output set $O$.
 2: Repeat 3-9 for $t$ iterations.
 3: **if** $O$ has less than $k$ elements **then**
 4:    Run Sample-Approx-K-Hot only with $p_i$ where $x_i \neq 1$ and $k = k - |O|$.
 5:    Add result to $O$.
 6: **else if** $O$ has more than $k$ elements **then**
 7:    Run Sample-Approx-K-Hot only with $p_i$ coordinates with $x_i = 1$, $k = |O| - k$ and $p = 1 - p$.
 8:    Remove result from $O$.
 9: **end if**
10: **Function** Sample-Approx-K-Hot $(p, k)$
11: Compute $s = \sum p_i$.
12: **if** $k <= s$ **then**
13:    Optionally apply correction to $kp_i/s$ for each $i$ to obtain corrected probabilities $p'_i$.
14:    $x_i = Bernoulli(p'_i)$ for each $i$.
15: **else**
16:    Optionally correct $(n-k)(1-p_i)/(n-s)$ for each $i$ to obtain corrected probabilities $p'_i$.
17:    $x_i = Bernoulli(p'_i)$ for each $i$.
18: **end if**
19: **if** $k > s$ **then**
20:    $x = 1 - x$.
21: **end if**
22: Return $x$
23: **EndFunction**

---

In the forward pass, we run the approximate conditional Poisson sampling procedure to obtain a $k$-hot sample $x$, which can be used as input to the downstream neural network. In the backward pass, the gradient relative to $x$ is used as the straight-through gradient relative to $q_i$.

As the new Poisson sampling procedure relies only on Bernoulli samples, we can also relax the entire approximate conditional Poisson sampling procedure by the Gumbel-Sigmoid relaxation (Maddison et al., 2017). In this paper, however, we rely exclusively on the described straight-through gradient estimation, since it showed to work well in our experiments. The full algorithm is described in Algorithm 3 and pseudocode is given in the appendix in Algorithm 4. Also, Figure 2 shows the modular architecture with the conditional Poisson subset sampling and straight-through gradients.

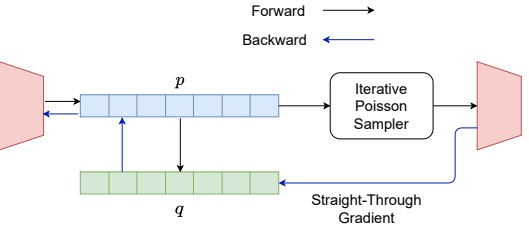

Figure 2: Diagram of the proposed $k$-hot sampling procedure with straight-through gradients. The vector of probabilities $p$ is output by a neural network. The vector $q$ is computed using Equation 1.

### 3.4 OTHER CONSIDERATIONS

**Number of iterations.** The method described in Algorithm 3 depends on repeating the Poisson sampling procedure for a number of iterations $t$. Here we note that $t$ does not depend on the subset size parameter $k$ unlike Top-$k$ procedures and is a fixed constant. Second, we note that the function of the parameter $t$ is to reduce variance in the subset size and a few iterations is enough to obtain with high probability a sample size within $\pm 1$ of $k$, even for large $k$. In our experiments, we choose $t$ between 5 and 8. Empirical results on the sample variance are discussed in Appendix E and Figure 1.

**Differentiable subset size.** We observe in Equation 1 and Algorithm 3 that the subset size parameter $k$ appears only as a scaling parameter in the probability vector. This implies that $k$ can be made differentiable assuming that the samples can be differentiated, for instance using the straight-through estimator as described earlier or some kind of relaxation. In this case, we learn $k$ as a fraction of the overall number of elements $n$, i.e. $k/n$ (as a continuous sigmoid output), and then rescale it by $n$. Note that this procedure is different from Top-$k$ procedures, which usually use $k$ in an internal loop index, making differentiation difficult.

**Correction for inclusion probabilities.** Given a set of desired probabilities $p_i$ such that $\sum p_i = k$, the conditional Poisson design does not necessarily lead to inclusion probabilities $\pi_i$ such that $\pi_i = p_i$. However, it gets approximately close, i.e. $\pi_i \approx p_i$, when $d := \sum_i p_i(1 - p_i)$ is large (Hájek & Dupač, 1981). Furthermore, Bondesson et al. (2006); Lundquist (2009) suggest corrections for the sampling probabilities to improve the approximation, which we discuss in more detail in Appendix C.

**Complexity.** The parallel (vectorized) complexity of the method (Algorithm 3) is constant, $O(1)$, up to the logarithmic factors required for reduction operations such as summation. This is because each

Poisson sampling step has constant parallel complexity up to reductions and we only iterate for a constant number of steps because of exponential variance reduction.

## 4 RELATED WORK

A few approaches for selecting subsets have appeared in the literature. Some of these methods (Pietruszka et al., 2021; Plötz & Roth, 2018; Xie et al., 2020) are designed to satisfy pre-defined constraints, such as fixed subset size. Other methods use regularization objectives to find some optimal subset size (De Cao et al., 2020; Louizos et al., 2018). The methods can be further classified in terms of whether they are deterministic (Pietruszka et al., 2021; Plötz & Roth, 2018; Xie et al., 2020) or stochastic (Chen et al., 2018; Paulus et al., 2020; Xie & Ermon, 2019).

Deterministic methods with constraints on the subset size often depend on relaxations of the Top-$k$ operator for selecting the largest $k$ elements from a set. One such relaxation is developed by Plötz & Roth (2018) as a repeated temperature-scaled softmax that is iterated $k$ times. At each iteration, an element is sampled from the categorical distribution induced by the softmax. For the next iteration, the categorical distribution is re-normalized after setting the probability of the selected sample to zero. The relaxation replaces samples from the distribution by expectations. Top-$k$ relaxations have been proposed with optimal transport (Xie et al., 2020) and tournament selection (Pietruszka et al., 2021).

The Top-$k$ relaxations are deterministic operations which can be combined with ranking distributions for sampling subsets. Building on Reservoir Sampling Efraimidis & Spirakis (2006), Xie & Ermon (2019) define a subset sampling operation by using the Gumbel distribution as a ranking distribution and using the Top-$k$ relaxation defined by Plötz & Roth (2018). A similar approach is taken by Goyal et al. (2018). Another sampling method is used by Chen et al. (2018) where independent samples are taken from the Concrete distribution (Maddison et al., 2017) followed by the element-wise maximum. This has the disadvantage that features may be repeated and fewer than $k$ elements might be finally selected.

In comparison, our proposed method is stochastic, and we impose a weaker constraint on the expected subset size rather than a hard constraint on the exact size and reduce variance. We also allow controlling the subset size per-instance by adding a regularization objective.

Poisson sampling, conditional Poisson sampling (Hájek & Dupač, 1981; Tillé, 2006) and related methods have been traditionally used in applications such as survey design (Ogus & Clark, 1971) and the consumer price index (Ohlsson, 1990).

## 5 EXPERIMENTS

We validate neural conditional Poisson sampling on three different tasks: model explainability for text and image classification Chen et al. (2018), high resolution image sub-sampling Huijben et al. (2019), as well as differentiable $k$-nearest neighbor search Plötz & Roth (2018). Specifically, image sub-sampling requires sampling from as many as 260K elements, which is the pixel resolution of the images. Sampling from so large spaces is intractable with current models due to extreme dimensionality, showing the scalability of the proposed method. Last, we validate with ablation experiments the efficiency, variance reduction, and inclusion probability approximation for the method. For lack of space, the latter experiments can be found in Appendix E.

### 5.1 LEARNING TO EXPLAIN TEXT CLASSIFICATION

For the task of model explainability, we work with the Learning to Explain framework of Chen et al. (2018). The aim is to generate *post hoc* instance-wise explanations of a classifier model, achieved by building an explainer network $e$ to select the $k$ input features with maximal mutual information with the model prediction per instance.

Since it is difficult to compute the mutual information directly, the framework works with a variational lowerbound of mutual information parameterized by a neural network. The optimization problem solved by in the learning to explain framework is written as

$$\max_{e,q} \mathbb{E}[\log q(X_S)], \text{ where } S \sim e(X), \qquad (2)$$

Table 1: Learning to explain text classification measured in *post-hoc* accuracy.

| Model | IMDB ($k = 10$) | 20NewsGroup ($k = 25$) | 20NewsGroup ($k = 50$) |
|---|---|---|---|
| L2X (Chen et al., 2018) | 90.8 | 50.5 | 34.7 |
| RSS (Xie & Ermon, 2019) | **91.7** | 58.6 | 58.0 |
| RSS (Xie & Ermon, 2019) + ST | **91.7** | 51.9 | 58.4 |
| NCPSS | 91.2 | **67.9** | 66.5 |
| NCPSS + differentiable $k$ | 91.2 | 65.7 | **68.2** |

Table 2: Learning to explain image classification measured in post-hoc accuracy. The RSS baseline runs out of GPU memory on sizes greater than 100. The baseline CIFAR-10 and STL-10 models have 80% and 75% accuracy respectively

| | CIFAR-10 | | | STL-10 | | | |
|---|---|---|---|---|---|---|---|
| $k$ | 50 | 100 | 150 | 100 | 400 | 600 | 700 |
| RSS (Xie & Ermon, 2019) | 50.7 | 54.1 | – | 42.5 | – | – | – |
| NCPSS | **63.6** | **65.2** | 68.0 | **50.6** | 56.3 | 57.6 | 59.6 |
| NCPSS + differentiable $k$ | 60.3 | 64.9 | 65.3 | 46.2 | 50.1 | 51.7 | 53.4 |

where $q(X_S)$ is the variational lower bound parameterized by a neural network, $X_S$ is the subset of $k$ features output by the explainer network $e$.

In practice, the output of the explainer network is a $k$-hot vector distribution from which a sample $S$ is used to mask features of the true input $X$ by element-wise multiplication, i.e., $X_S = X \odot S$. For the text classification experiments we use the Large Movie Review Dataset (Maas et al., 2011) with two classes for sentiment classification. We follow the same pre-processing as in Chen et al. (2018) by resizing each review to 400 words. We also use the 20Newsgroups dataset (Rennie & Lang, 2008) for classification of posts in 20 different newsgroups. We resize each document to 1000 words by padding or cutting.

The models to be explained in both cases are convolutional neural networks, which achieve 90% and 70% test set accuracy on IMDB and 20Newgsroups, respectively. For the 20Newsgroups dataset we use pretrained GLoVe embeddings (Pennington et al., 2014), while for IMDB we train from scratch. For the IMDB dataset we select $k = 10$ words as explanations, while for the 20Newsgroups dataset we use $k \in \{25, 50\}$ words. We use the same network architecture as in Chen et al. (2018) for the IMDB dataset. For 20Newsgroups we use a similar architecture but with two extra convolutional layer each in the explainer and the variational network and a filter size of 128. We evaluate the final performance using *post hoc* accuracy (Chen et al., 2018). For evaluation we always use *hard* binary vectors as masks $S$.

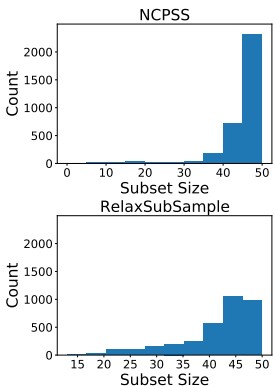

The results are shown in Table 1. In the simpler IMDB dataset both our model and RelaxSubSampling (RSS) can explain well classifications with a small number of words $k$, matching the accuracy of the full model. For the larger 20Newsgroups, however, neural conditional Poisson sampling outperforms RelaxSubSampling by a significant margin of 7-10%, while coming close to the accuracy of the full model. Moverover, when comparing sampled tokens, we see that RelaxSubSample (Xie & Ermon, 2019) samples often invalid words like padding tokens unlike neural conditional Poisson sampling, as seen in Figure 3. This shows that RelaxSubSampling is unable to take advantage of larger subset sizes when going from $k = 25$ to $k = 50$. These results validate our claim that our method performs better with larger subsets,

Figure 3: Histograms showing frequency of sampled tokens that correspond to actual words and not padding for our method (top) and RelaxSubSample (bottom) on the 20Newsgroups evaluation set. $k$ is set to 50 and only documents with at least 50 words were considered. RelaxSubSample shows a large variation in the number of actual words selected.

whereas Gumbel Top-$k$ relaxations deteriorate because of their longer softmax iteration chains. For NCPSS, both $k = 25$ and $k = 50$ achieve post-hoc accuracies close to the original test accuracy.

edu writes stuff del for bandwidth sake why sigh if you don have more than mbs of memory ==using== with ==windows== is of ==memory== ==windows== will access mb ram better as memory as to why what you did didn work it is because and paths are stored inside the group ==ini== ==files== all of the sudden things went from drive to drive however if you wanted to copy an application up to the and re setup it up that should work normally but as previously stated this will only hurt things unless you ve got more than mbs of ==ram== and are using whats above as the personally have mb of ==ram== and run ==mb== with great deal of success however if you are looking to speed up ==windows== the three things ve noted that work the best are ==graphics== card ==co== processor even an ==helps== some other ==disk== ==cache== besides ve tried several and lightning for ==windows== and norton cache give me major headaches as well think the purpose the original poster was trying to serve is to avoid the significant amount of ==disk== access that ==windows== does on ==startup== it like it trying to it bit in wearing the damn drive out estimate it only reading ==mb== of programs data but from the performance the drive gives it sounds like they are scattered all over the drive my drive is however compressed what is it that takes so much perhaps if ms would take the trouble to this startup process less people would be wanting to find solution themselves

Class: comp.os.ms-windows.misc, Predicted: comp.os.ms-windows.misc, Words Selected: 20/261, Total Tokens Selected: 20

Figure 4: Example text explanation with the 20Newsgroups dataset for a correctly classified document with differentiable set size with an average explanation size of 50 words. In this case the model chooses a smaller explanation size of 20 words. In this case all chosen tokens correspond to actual words and no padding tokens or similar are selected. See the appendix for examples on longer inputs.

**Differentiable explanation sizes.** For explaining text documents with 20Newsgroups we also experiment with learning optimal differentiable subset sizes of $k$, for which we consider a prior $\hat{k}$ to be either 25 or 50. For this we add a squared loss term in the loss expression as $\gamma(\mu_k - \hat{k})^2$, where $\mu_k$ is the mini-batch average $k$ computed the network, and $\gamma$ is the regularization strength chosen from $\{0.1, 0.01, 0.001\}$. Constraining only the average explanation size allows the model to choose the explanation size per instance, often yielding even stronger explanations in terms of *post-hoc* accuracies, see Table 1. For instance, in Figure 4 the model chose a 20-word explanation for a correct classification although the average explanation is conditioned to 50 words. Examples of longer explanations can be found in the appendix.

## 5.2 LEARNING TO EXPLAIN IMAGE CLASSIFICATION

We repeat the experiment now for explaining image classification models on CIFAR-10 (Krizhevsky, 2009) and STL-10 (Coates et al., 2011) using the same "learning to explain" framework (Chen et al., 2018) and sub-pixel explainers. This means that the explainer $e$ outputs a 3x32x32 size mask for CIFAR-10 with 32x32 resolution, and 3x96x96 for STL-10. Generating such explanations requires large subset sizes. For CIFAR-10 we explain a simple CNN model with 8 convolutional layers that achieves 80% val. accuracy, and for STL-10 a ResNet-10 model that achieves 75% val. accuracy.

We choose a simple CNN explainer and variational network architecture. For CIFAR-10 the explainer has 5 convolutional layers of 64 filters and no subsampling. The variational network has 3 convolutional layers of size 32 with 4x4 max pooling layers after each intermediate layer and 2x2 average pooling at the output. For STL-10 the explainer has 5 convolutional layers with 96 filters. The variational network has 5 hidden convolutional layers of sizes $[64, 128, 256, 512]$ with max pooling layers for downsampling and a final global average pooling layer with 10 outputs.

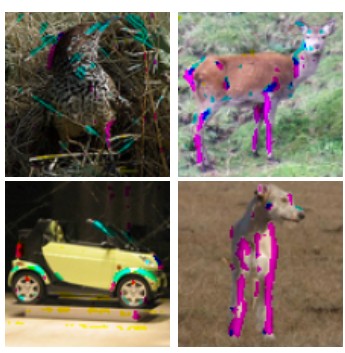

Figure 5: Learning to explain image classification with 700 sub-pixels on STL-10.

We compare against RelaxSubSample (Xie & Ermon, 2019) for $k = 50, 100$ for CIFAR-10, and $k = 100$ for STL10. For larger subset sizes, we experienced RelaxSubSample to run out of GPU memory. The reason is the large input sizes on STL-10, as well as the fact that RelaxSubSample needs to perform $k$ softmax operations per image, and all must be kept in memory for backpropagation. In contrast, NCPSS is easily scalable, such that we also run our method with $k \in \{400, 500, 600, 700\}$ on STL-10. We choose the relaxation temperatures for RelaxSubSample from $\{0.1, 0.5, 1.0\}$.

We show results in Table 2. Overall, neural conditional Poisson sampling outperforms the Gumbel-Top-$k$ RelaxSubSample by a large margin on the same subset sizes, over 10% for CIFAR-10 and 8% for STL-10. Furthermore, we see a clear increase of performance with larger subset size, indicating that NCPSS makes efficient use of the set sizes. Note that in these experiments, the *post hoc* accuracy of the best model is still below the original model, which can likely be addressed by a better explainer or variational network architecture. We provide examples explanations (the negative of the generated mask, for better illustration) of correct predictions by our method in Section 5.2. Finally, for CIFAR-10 and STL-10, we also experiment with differentiable subset sizes with our method when the average explanation size is regularized to be, e.g., 50 and 100 by adding a term in the loss function as we did for our document classification experiments. Generally, we find that adding a differentiable average subset size constraint leads to worse evaluation accuracy.

**Subset size variation.** We show that our method leads to very low variation in the sampled subset sizes on real data with only a few iterations $t$. We sample 1000 subsets for different input data points and compute the minimum, maximum, and mean subset size for $k \in \{50, 100\}$ and $t \in \{3, 5, 8\}$. Results in Table 8 in Appendix E show only a small variation ($\pm 4$ at $t = 8$) in subset sizes given $t$.

**Efficiency comparison.** We compare the time taken per epoch for our method and the baseline Gumbel Top-$k$ method RelaxSubSample for increasing subset size $k$ on the learning to explain task on CIFAR-10, see Table 7 in Appendix E. The proposed algorithm, NCPSS, trains almost twice as fast as RelaxSubSample for $k = 50$, and twice as fast for $k = 100$. In general, the runtime was only minimally affected by different subset sizes $k$ and sampling iterations $t$.

### 5.3 SUBSAMPLING LARGE IMAGES

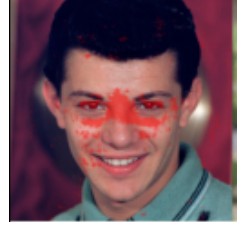
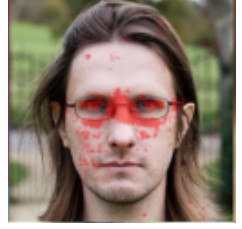

Last, to showcase the capacity of neural conditional Poisson sampling for large-scale inputs and experiments, we subsample large images for a downstream classification task following Huijben et al. (2019). For this, we create a subset of the CelebA-HQ (Lee et al., 2020) dataset with 512x512 images, half of them featuring the *eyeglasses* attribute and the other half not. The *eyeglasses* attribute then serves as the classification target. The task of the subsampler is to compute a global input mask. We replace the Gumbel sampling layer used by Huijben et al. (2019) with our iterative Poisson sampler. We subsample 5, 10 and 15 percent of the pixels which are fed to the downstream classifier. We use a small 6-layer CNN with max-pooling layers for downsampling, a final global average pooling layer for the output, and train the model for 80 epochs. We obtain accuracies of 93.4, 95.1 and 95.4 percent for 5, 10 and 15 percent pixels respectively. This experiment is intractable with the method from Huijben et al. (2019) due to high memory usage. Example images are shown in Figure 6. We also experimented with *per-instance* features for eye glasses classification. Qualitative results for this can be seen in Appendix G.1.

Figure 6: 512x512 sub-sampled Celeb HQ images for eye glasses classification. Selected pattern is shown in red.

By contrast, the method from Huijben et al. (2019) is infeasible for such large images since it attempts to create the full subsampling matrix, requiring over 27GB of memory for 512x512 images and batch size of 10. To make a comparison with Huijben et al. (2019) possible, we downscaled images to 64x64, on which NCPSS obtains 2-3% higher accuracy (91.1 v. 89.1 for 15% selected pixels and 89.4% v. 86.6 for 10% selected pixels, see Appendix D).

## 6 CONCLUSION

We presented a method for sampling a subset of $k$ elements from an $n$ element universe. The method is intermediate between Poisson and conditional Poisson sampling by reducing variance in Poisson sampling and generating samples that are close to $k$-hot. The main limitations of the method are: 1) the subset size $k$ is only achieved in expectation, so there is not much advantage gained for very small subsets in terms of complexity, and 2) there are no theoretical guarantees for the straight-through gradient. In spite of this, the method is efficient and scalable to sampling of large subset sizes beyond what is achievable by current Gumbel Top-$k$ methods.

## 7 ACKNOWLEDGEMENTS

This work is financially supported by Qualcomm Technologies Inc., the University of Amsterdam and the allowance Top consortia for Knowledge and Innovation (TKIs) from the Netherlands Ministry of Economic Affairs and Climate Policy.

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

## A    PSEUDOCODE FOR ITERATIVE POISSON SAMPLING

---

**Algorithm 4** Iterative Poisson Sampling Pseudocode

---

**Require:** Input $p \in \mathbb{R}^d$, $k$ integer, $t$ integer;
1: Set $i = 0$, $f = 1$, $S = 0^d$, $mask = 1^d$, $pin = p$, $kin = k$
2: **while** $i < t$ **do**
3:    $s = $ Sample-Approx-K-Hot$(p, k, mask)$
4:    Set $S = S + f \cdot s$
5:    Set $r = \sum S$
6:    **if** $k <= r$ **then**
7:       Set $f = -1$, $mask = s$, $p = 1 - pin$, $k = r - kin$
8:    **else**
9:       Set $f = 1$, $mask = 1 - s$, $p = pin$, $k = kin - r$
10:   **end if**
11: **end while**

12: **Function**    Sample-Approx-K-Hot    $(p, k, mask)$
13: Set $c = \sum p \cdot mask$, $n = \sum mask$
14: **if** $k <= c$ **then**
15:    Set $q = kp/c$
16: **else**
17:    Set $q = (n - k)(1 - p)/(n - c)$
18: **end if**
19: Sample $s$ independently from $q$
20: **if** $k <= c$ **then**
21:    Return $s \cdot mask$
22: **else**
23:    Return $(1 - s) \cdot mask$
24: **end if**
25: **EndFunction**

---

## B    PROOF OF PROPOSITION 3.1

**Proposition 3.1.** *Let $S_i \in \{0, 1\}^n$ denote the subset and $q_i$ bounding the probability at step $i$ in the iterative Poisson sampling procedure. Then after $T$ iterations we have $Var(|S_T|) \leq Var(|S_1|) \prod_{i=2}^{T} q_i$, where $Var(|S_1|)$ is the Poisson sampling variance. Assuming that the probability bound is bounded away from 1, i.e., $q_i < 1 - \epsilon$ for $\epsilon > 0$, the procedure obtains an exponential decrease in variance with the number of steps $T$.*

*Proof.* By design we have that $\mathbb{E}[|S_i| \mid S_{i-1}] = k$, where $k$ is the required subset size. Note that $Var(|S_1|)$ is the plain Poisson sampling subset size variance. Using the following upper bound for the variance of the binomial distribution,

$$Var(; n, p) = np(1 - p) \leq np,$$

we have

$$Var(|S_i| | S_{1:i-1}) \leq (n - |S_{i-1}|)(k - |S_{i-1}|)q_i.$$

Combining with $\mathbb{E}[X^2] = Var(X) + \mathbb{E}[X]^2$, we get

$$Var(|S_i| \mid S_{1:i-2}) = \mathbb{E}[Var(|S_i| \mid S_{1:i-1})] \tag{3}$$

$$\leq \mathbb{E}\left[nk - n|S_{i-1}| - k|S_{i-1}| + |S_{i-1}|^2 \mid S_{1:i-2}\right] q_i \tag{4}$$

$$\leq q_i Var(|S_{i-1}| \mid S_{1:i-2}). \tag{5}$$

Unrolling from $T$ to 1 we get

$$Var(|S_T|) \leq Var(|S_1|) \prod_{i=2}^{T} q_i.$$

Combined with the assumption that $q_i$ are bounded away from 1, the right hand side gives an exponential decay in the Poisson sampling variance $Var(S_1)$ over $T$ steps.    □

Conditional Poisson sampling would need on average $1/\binom{n}{k} p^k (1 - p)^{n-k}$ steps given that the success probability for size $k$ in each sample is $\binom{n}{k} p^k (1 - p)^{n-k}$.

## C    Correction for Inclusion Probability

Given desired probabilities $p_i$ such that $\sum p_i = k$, the conditional Poisson design does not lead to inclusion probabilities $\pi_i$ such that $\pi_i = p_i$. However, $\pi_i \approx p_i$ and the approximation improves when $d := \sum_i p_i(1 - p_i)$ is large Hájek & Dupač (1981). Bondesson et al. (2006); Lundquist (2009) develop corrections $p_i'$ for sampling probabilities to improve the approximation relative to the desired inclusion probabilities $p_i$ depending on whether $d$ is large or small. For large $d$, the suggested correction uses $p_i'$ such that

$$\frac{p_i'}{1 - p_i'} = \alpha \frac{p_i}{1 - p_i} \exp\left(\frac{1/2 - p_i}{d}\right), \tag{6}$$

where $\alpha$ is chosen so that $\sum_i p_i' = k$. Another correction that subsumes the large $d$ and very small $d$ cases is to use $p_i'$ such that

$$\frac{p_i'}{1 - p_i'} = \alpha \frac{p_i}{1 - p_i} \exp\left(a \operatorname{arcsinh}\left(\frac{1/2 - p_i}{a \cdot d}\right)\right), \tag{7}$$

where $a$ is recommended by Lundquist (2009) to be chosen as $a = 1/2 + d^3/2$. For our experiments we found the correction from equation (7) work better than the one from equation (6), which we choose to use. In general, in our experiments, we find the corrections to be useful only when $n$ is small and $k$ is less than $n/2$. Otherwise, for large $n$ the inclusion probabilities were approximated quite well by the uncorrected probabilities $p_i$ in our experiments. We experimentally validate the inclusion probability correction in Appendix E.

## D    Subsampling Comparison

We compare the performance of our method on an image subsampling task for downstream classification with deep probabilistic sampling (DPS) Huijben et al. (2019) which uses Gumbel sampling for sampling subsets. We use the CelebA dataset for this experiment. Since we find that the DPS method is infeasible to use on large images due to its high memory requirement, we compare on images downscaled to 64x64. We prepare the dataset as described in Section 5.3 and build a classifier for the eyeglasses attribute. We use a 5 layer CNN with maxpooling downsampling layers and train for 100 epochs. For DPS we use temperatures of 0.5 and 1.

Table 3: Subsampling comparison on CelebA eyeglasses attribute classification evaluation set

| Model | Percent Pixels | Percent Evaluation Accuracy |
|---|---|---|
| NCPSS | 10 | 89.4 |
| DPS Huijben et al. (2019) | 10 | 86.6 |
| NCPSS | 15 | 91.1 |
| DPS Huijben et al. (2019) | 15 | 89.1 |

Table 4: Subsampling performance on CelebA-HQ 512x512 images for eyeglasses attribute classification on the evaluation set

| Model | Percent Pixels | Percent Evaluation Accuracy |
|---|---|---|
| NCPSS | 5 | 93.1 |
| NCPSS | 10 | 95.1 |
| NCPSS | 15 | 95.4 |

### D.1    Timing Comparison

Our method is significantly more efficient than Top-$k$ methods for large $k$. Below we provide timing comparison for the subsampled eyeglasses classification experiment described above with 64x64 images. The times are for 80 epochs.

Table 5: Training time comparison on CelebA 64x64 images for subsampled eyeglasses attribute classification

| Method | Pixels | Time/80 epochs |
|---|---|---|
| NCPSS | 10% | 41 minutes |
| DPS Huijben et al. (2019) | 10% | 3 hours |
| NCPSS | 15% | 41 minutes |
| DPS Huijben et al. (2019) | 15% | 4.4 hours |

## E   FURTHER VALIDATION OF THE METHOD

In this section we verify some properties of the proposed method. First, we verify that the actual inclusion probabilities are close to the desired probabilities and the conditions under which the correction described in Section C improves the approximation. Second, we verify that the method indeed reduces variance in the subset size at the exponential rate suggested by Proposition 3.1.

### E.1   VERIFYING THE APPROXIMATION FOR INCLUSION PROBABILITIES

First we verify that our proposed method generates $k$-hot samples with probabilities close to the prescribed probabilities. We generate random probability vectors, $p$, of dimension chosen from $\{100, 500\}$ that are normalized to sum to 1. Next we generate 1000 $k$-hot vectors using our method both with and without the correction described in Appendix C. We estimate the inclusion probability by averaging across the 1000 $k$-hot samples for each dimension, denoting the empirical probabilities by $\hat{p}$. We compute the mean squared error as $\sum_i (p_i - \hat{p}_i/k)^2$. We repeat the procedure for values of $k \in \{10, 20, 30, 40, 50, 60\}$. The root mean squared (RMS) error with and without the correction for various values of $k$ is shown in Table 6.

Table 6: Root mean squared error for inclusion probabilities

| | $k$ ($n = 100$) | | | | | |
|---|---|---|---|---|---|---|
| Method | 10 | 20 | 30 | 40 | 50 | 60 |
| NCPSS | 0.0165 | 0.0095 | 0.0071 | 0.0042 | 0.0030 | 0.0043 |
| NCPSS+cr. | 0.0116 | 0.0082 | 0.0060 | 0.0037 | 0.0027 | 0.0043 |

| | $k$ ($n = 500$) | | | | | |
|---|---|---|---|---|---|---|
| Method | 50 | 100 | 150 | 200 | 250 | 300 |
| NCPSS | 0.0048 | 0.0030 | 0.0021 | 0.0015 | 0.0017 | 0.0082 |
| NCPSS+cr. | 0.0045 | 0.0028 | 0.0020 | 0.0015 | 0.0017 | 0.0369 |

From the table we find that the empirical inclusion probabilities are already quite close to the desired probabilities without the correction for the chosen values of $n$ and $k$. The largest RMS error is about 0.016 for $k = 10$ and $n = 100$ and the error generally decreases for large values of $k$ up to $n/2$. We also find from the table that the correction helps more for smaller $n$ but only for values of $k$ less than about $n/2$ beyond which it makes the error worse. Since the results show that the correction can help improve the approximation when $k$ is small relative to $n$, we treat the correction as a modeling choice depending on the experiment.

### E.2   SUBSET SIZE VARIANCE

We examine the variance of the size of the subset selected with our method and its relation with the number of iterations $t$ performed by Algorithm 3. We choose a probability vector of $n = 3000$ dimensions and choose 1000 subsets of size $k = 300$ and run the algorithm for $t = 1$ and $t = 5$ iterations. Histograms of the subset size distribution are shown in Figure 1. The results show a significant decrease in variance with a small number of iterations.

### E.3 Efficiency Comparison

Table 7: Time per epoch in seconds versus $k$ on CIFAR-10

| Method | 50 | 75 | 100 | 150 | 200 | Baseline 1 |
|---|---|---|---|---|---|---|
| | | | $k$ | | | Baseline |
| RelaxSubSample | 22s | 28s | 32s | - | - | 10s |
| NCPSS | 13s | 13s | 13s | 13s | 13s | 10s |

### E.4 Subset Size Variation

Table 8: Variation in subset size with number of iterations $t$ on CIFAR-10.

| | $k = 50$ | | | $k = 100$ | | |
|---|---|---|---|---|---|---|
| $t$ | Mean | Min | Max | Mean | Min | Max |
| 3 | 49.9 | 45 | 60 | 99.7 | 94 | 105 |
| 5 | 49.95 | 47 | 54 | 100.1 | 95 | 104 |
| 8 | 50.01 | 49 | 52 | 99.9 | 96 | 102 |

### E.5 kNN Classification

We show that our method is also competitive against other methods when using smaller subset sizes ($k$=9) on a stochastic $k$ nearest neighbors classification task with deep features Grover et al. (2018); Xie & Ermon (2019). We use the same setup as Xie et al. (2020) except that we use cosine distance rather than Euclidean distance, since our method requires the weights of the elements to be normalized to $(0, 1)$.

We use $k = 9$ neighbours and compare against RelaxSubSample, NeuralSort Grover et al. (2018), SOFT Top-$k$ Xie et al. (2020). The results are shown in Table 9. From the result we see that although we do slightly worse on MNIST, we outperform all baselines except the deterministic SOFT Top-$k$ and are on-par with SOFT Top-$k$ on CIFAR-10. The results show that our method performs well and on-par with other similar methods, also in the regime of small subset sizes $k$.

Table 9: kNN Test Set Classification Accuracy for $k = 9$

| Model | MNIST | CIFAR-10 |
|---|---|---|
| kNN Grover et al. (2018) | 97.2 | 35.4 |
| kNN+PCA Grover et al. (2018) | 97.6 | 40.9 |
| kNN+AE Grover et al. (2018) | 97.6 | 44.2 |
| kNN+RelaxSubSample Xie & Ermon (2019) | 99.3 | 90.1 |
| kNN+NeuralSort Grover et al. (2018) | 99.5 | 90.7 |
| kNN+k-Softmax Xie et al. (2020) | 99.3 | 92.2 |
| kNN+Soft-Topk Xie et al. (2020) | 99.4 | 92.6 |
| kNN+NCPSS | 99.2 | 92.5 |

## F Learning to Explain: Text Examples

In the following we show examples of explanations generated by our method for the 20Newsgroups dataset with differentiable set size. The mean size was set to 50 words. This allows the model to choose per-instance explanation sizes.

in article liu se writes intersection between line and polygon by dave tom from graphics cornell edu in recent years many geometric problems have been successfully in new language called postscript see postscript language by adobe systems incorporated isbn co so given line and polygon we can write postscript program that draws the line and the polygon and then outputs the answer by output we mean the program command called which actually prints page of paper containing the line and the polygon quick examination of the paper provides an answer to the reduced problem and thus the original problem in modern postscript the point in polygon problem can be solved even more easily to wit title point in polygon creator allen ab cc purdue edu for the of comp graphics humor sense thereof this program will test whether point is inside given polygon currently it uses the even odd rule but that can be changed by replacing with these are level operators so if you ve only got level you re out of luck the result will be printed on the output stream caution only accurate to device pixels put huge scale in first if you aren sure point to test put and here of polygon in counter order put array of pairs of here get pop length roll sub pop yes no

Class: comp.graphics, Predicted: comp.graphics, Words Selected: 13/223, Total Token Selected: 13

in article apr sps mot com email sps mot com writes their problem wasn giving them any more money the finance guy then brought in the manager on duty who proceeded to give me hard time reminded him that was the customer and didn think should be treated like that and that if he didn back off he could forget the whole deal he made some smart remark so told him where he could stick it back my check and left needless to say they were not pleased by the turn of events that nothing when friend of mine went shopping for small sedan few years ago she brought me along as token male so the wouldn give her the treatment her first choice was mazda and second choice was nissan we went to mazda dealership and described what we wanted we started negotiating on the price and the kept playing the let me run this price by the sales manager after playing the good salesman bad salesman game we finally told him that if he didn have the authority to negotiate price perhaps we should be speaking directly to someone who did he brought in the sales manager who proceeded to dick us around with every trick in the book read don get taken every time for list finally after playing few more rounds of you ll have to work with us on this price also known as each time you come up thousand dollars we ll come down ten the gave signal to his two sales stood up and said well we can come down any more so guess we can help you and they out of the room leaving us sitting in the salesman office all by ourselves hmm read that sometimes bug their own offices so they can leave and listen in on discussing the sales offer and mentioned this to my friend while we were sitting there wondering why they would leave us in the office instead of showing us to the door for lack of anything better to do picked up the phone on the desk and called another mazda dealership asked for salesman and began discussing what kind of price they would consider few sentences into the conversation mr broke into the line and began telling me how rude he thought it was that would call another dealership from his phone said that since he announced that our business was over he shouldn care and every time tried to talk to the other sales guy the sales manager would out our voices with his own how did he know that was using the phone anyway finally hung up and we headed out of the sales manager and come out of little room and he begins to us again we say that we won bother him anymore we re going next door to the nissan dealership then comes the part wish could have as we go out the front door the sales manager across the entire customers and all go ahead you deserve to buy nissan so my friend bought just so the guilty won go ll mention that the sales manager name was gary from his manner his refusal to come down to reasonable price and his anger at the end my guess is that he had bet our original salesman who was young that he be able to get at least dollars out of us and he was that we wouldn fall for his tricks

Class: rec.autos, Predicted: rec.autos, Words Selected: 62/583, Total Token Selected: 64

previously wrote yeah the phillies played over their heads almost the whole year but it all caught up to them in one game streak am as old as man and was big phillies fan at the time age september is still painful thing to remember but can tell you that the phillies never led the league by that year going by memory alone believe their biggest lead was games and they were ahead when the famous game losing streak began streak during which it seemed that they found just about every way to lose known to man anyway think they just before the end and won their last couple games and were still in the thing until the final day but finished tied with the giants one game out and didn the dodgers or somebody else finish two games back that has to be one of the closest last minute ever ok you guys up my childhood memories so went and did some research on the final month or so of the season it turns out that my were pretty darn accurate at least as far as the phillies record goes on september this was the top of the standings gb philadelphia cincinnati st louis san francisco this is game by game description of the remainder of the phillies season date score opponent lead pitcher starting and winner loser houston houston short houston san francisco win san francisco san francisco short los angeles los angeles wise los angeles st louis cardinals take over nd place from cincinnati st louis short san francisco san francisco giants move into tie for nd with st louis san francisco cardinals back in sole of nd place houston short houston houston los angeles wise los angeles short los angeles los angeles reds move back into tie for nd with cardinals well so far so good for the phillies but now it all falls apart cincinnati reds take sole of nd place cincinnati short cincinnati milwaukee milwaukee short cards now back in rd giants in th milwaukee milwaukee lose games in days reds take over st cardinals back in rd st louis short cardinals take over nd place drop to rd st louis reds and cardinals now tied for st st louis cardinals take game lead over reds cincinnati short phillies game losing streak cards lead reds by game did not play cards lose to mets reds tied for st game back cincinnati cards beat mets take first by from reds and phillies what finish and the final standings were gb st louis philadelphia cincinnati san francisco now it doesn appear to me that phillies pitchers and short were really at least by the four man rotation standard of the day until well along into the game losing streak at which time was probably desperate for win at any cost because the phillies substantial lead had the way they were used at that time may have made the problem worse although had one of his games of the year in the final day of the reds that cost the reds share of the pennant pitched complete game six hitter striking out five and walking one it would be to see though how the total innings for the year for and short up against the rest of the league also notice that the phillies played every day from at least september through october while they didn play substantially more games than the other teams the other teams each had couple days off during that stretch eric smith netcom com com ci

Class: rec.sport.baseball, Predicted: rec.sport.baseball, Words Selected: 77/594, Total Token Selected: 81

in article ponder jesse writes hi have you used mac system or if the answer is positive you would know if ms windows is mature os this is silly is unix mature os depends on who you ask and how you define mature system is if anything less mature than windows days ago people that ms windows is not real os can see why they have such question ms windows many people microsoft mac but it did lousy job for example you can not create hierarchy groups there is no way to create group in group if you know how please tell me so why do you need something like to create groups under the apple menu everyone knows that apple menu items are of the program manager if you want program launcher there are lots available documentation it not easy to find the reason why causes an error and this is easy on mac give me break having spent hours moving system extensions around and the mac to see why certain app crashes all the time find this group file after group users have to use file manager to delete files but if users forget to delete some related files the disk will be full of nonsense files oh great ever hear of wonder why apple implemented them share problem once you create two windows doing and editing in some language good editor there will be sharing problem you just can not open or save the program if it is loaded it makes sense to prevent from saving but not opening eh don follow it by no means easy to satisfy everybody but if microsoft want to keep their they should evaluate the user interface more carefully before products distribute why is it that find the mac desktop incredibly annoying whenever use it no flame please yeah right you post flame bait yet ask for no flames the only thing worse than ala internet silver ucs indiana edu frog is frog ala bitnet

Class: comp.os.ms-windows.misc, Predicted: comp.os.ms-windows.misc, Words Selected: 27/334, Total Tokens Selected: 27

---

am very interested in hearing from all of you who are using or implementing interactive applications what types of widgets you would like to have in your applications widget is usually located in the same scene as other objects of the application it may let you manipulate application data the camera objects in the scene and so on or view the status of the application or objects via the widget shape color position orientation and so on or do whatever missed but you think is possible for example widget can be virtual shown as partially transparent sphere super imposed on the object to be feedback widget can be with ends to objects the length of the changes as the objects move and value is shown on the indicating the distance widget can provide both manipulation and feedback for example the can be used to change the distance between the objects along its own axis please mail me or post your opinions on interaction the information gathered will help me design ui construction tool your help is very much appreciated tony lau cs ubc ca sc student dept of computer science

Class: comp.graphics, Predicted: comp.windows.x, Words Selected: 44/190 Total Token Selected: 48

# G    IMAGE CLASSIFICATION EXAMPLES

## G.1    CELEBA

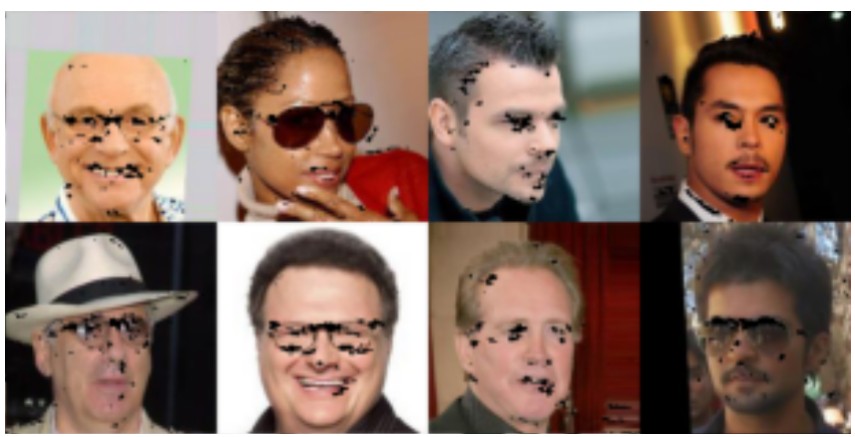

Figure 7: Per-instance examples on Celeba with 150x150 images for eye glasses attribute classification with 500 pixels.

## G.2    CIFAR-10

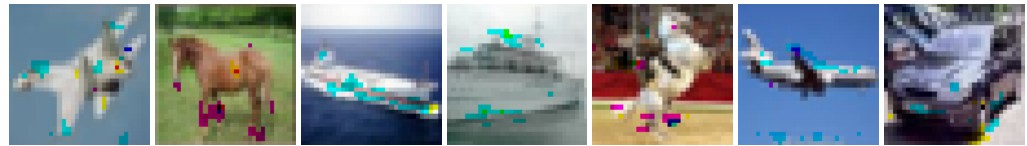

Figure 8: Learning to explain image classification with 100 sub-pixels on CIFAR-10.

## G.3    STL-10

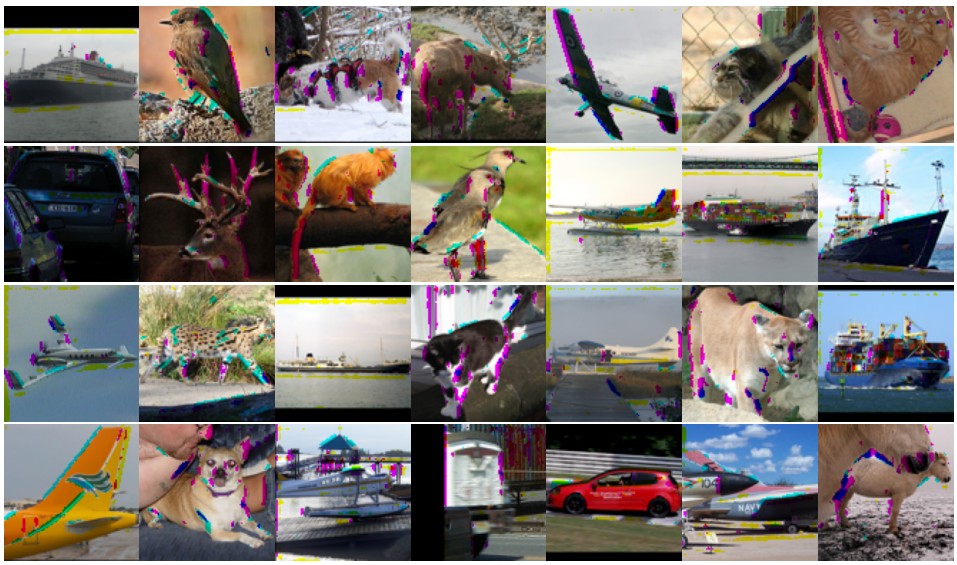

Figure 9: Learning to explain image classification with 700 sub-pixels on STL-10.

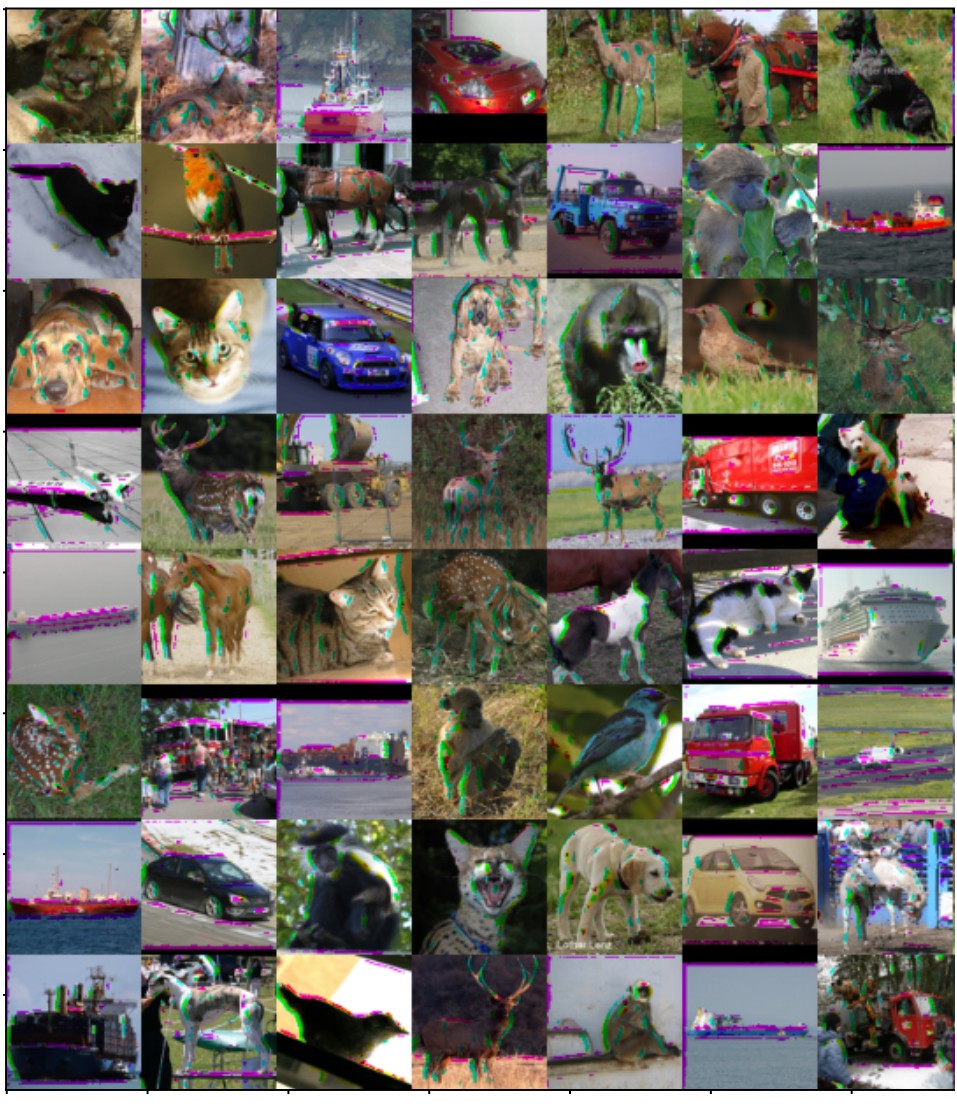

Figure 10: Learning to explain image classification with 800 sub-pixels on STL-10.

