# OpenReview forum: "Scalable Subset Sampling with Neural Conditional Poisson Networks"
_ICLR.cc/2023/Conference — ICLR 2023 poster_

### Official Review · Reviewer_ZXrw · 2022-10-24

**Confidence:** 4
**Correctness:** 4
**Technical Novelty And Significance:** 3
**Empirical Novelty And Significance:** 2
**Recommendation:** 8

**Clarity, Quality, Novelty And Reproducibility:**

This paper is reasonably high quality, well written, and clear.  While relatively straightforward, the proposed approach appears to be original.  Enough detail is included in the paper to implement and reproduce the proposed approach, and code is provided in order to reproduce the experimental results.

**Strength And Weaknesses:**

Strengths:
* The proposed approach is scalable, allowing for efficient sampling of large subsets.
* The proposed approach is fully differentiable, allowing it to be incorporated into differentiable models that are trained via gradient-based learning methods.
* The experiments in this paper show strong empirical evidence that the proposed approach outperforms competing approaches in terms of both predictive performance and runtime performance, sometimes by a significant margin.

Weaknesses:
* The proposed neural conditional Poisson subset sampling assumes that the $k$ elements for a sampled subset are drawn IID from the universe of $n$ elements.  That is, it assumes that there are no interactions (correlations) between elements within a subset.  This is a significant limitation.  Scenarios involving text or image data may often involve interactions between elements within a subset, which may not be represented well with the approach in this paper.
* No theoretical analysis of the runtime complexity of the proposed algorithm (Algorithm 3) is provided in the paper.


**Summary Of The Paper:**

This paper describes a method for subset sampling based on conditional Poisson sampling, called neural conditional Poisson subset sampling.  This approach builds on the general conditional Poisson sampling approach, which samples each element in the subset independently, and conditions this procedure to return subsets of exactly $k$ elements.  Conditional Poisson sampling has some weaknesses, including high computational complexity, and that it is not differentiable and thus cannot be included in differentiable models.  The proposed neural conditional Poisson subset sampling approach addresses these issues, allowing for efficient sampling of large subsets, and is fully differentiable.

**Summary Of The Review:**

While the approach described in this paper is relatively simple and straightforward, it is in fact more scalable than a number of prior approaches for subset sampling, and also has the advantage of being fully differentiable.  This paper is well written, and the empirical results are convincing.  Taken together, this is a good paper and should be accepted.

---

> ### Author Response · Authors · 2022-11-12
> **Author Reply**
>
> We thank the reviewer for reviewing our work.
>
> >>*Weaknesses:*
> >>
> >>    *The proposed neural conditional Poisson subset sampling assumes that the $k$  elements for a sampled subset are drawn IID from the universe of elements. That is, it assumes that there are no interactions (correlations) between elements within a subset. This is a significant limitation. Scenarios involving text or image data may often involve interactions between elements within a subset, which may not be represented well with the approach in this paper.*
> >>
>
> IID sampling is only true for one step of Poisson sampling. However, when sampling a subset  of size $k$ we will get some correlations between elements. Similarly for our case, because of the multiple iterations, we do get some correlations between elements.
>
> Nevertheless, the subset assumption, in general, is indeed a limitation of the framework for generating explanations that treats feature vectors as a set. Since choosing an element does not change the probability that related elements will also be chosen.
> However, this affects all methods that use this framework including the baselines from prior work (such as L2X and Gumbel Top-$k$ methods) and is not specific to our method.
> This limitation is somewhat mitigated by fact that the weights for the elements are learned by deep networks which can learn correlations between various elements.
>
> >>*No theoretical analysis of the runtime complexity of the proposed algorithm (Algorithm 3) is provided in the paper.*
> >>
>
> The parallel (vectorized) complexity of our method is constant up to logarithmic factors for reduction operations such as summations, or computing the minimum or maximum.
> In particular, it does not depend on the size of the universe $n$ or the subset size $k$.
> Each Poisson sampling step has constant parallel (vectorized) complexity because of independent sampling up to logarithmic factors.
> Due to exponential variance reduction (Proposition 3.1 and Figure 1) we only repeat the procedure a constant number of (no more than 8 to 10) steps.
> Taken together this implies constant parallel complexity, $O(1)$, up to the cost of reduction operations.
> We clarify this in Section 3.4 in the updated draft.
>
> --
>
> We hope to have addressed the reviewer's concerns. If there are other issues which require clarification, please let us know.

---

> > ### Comment · Reviewer_ZXrw · 2022-11-15
> > **Rebuttal response**
> >
> > I thank the authors for their rebuttal comments and explanations. I have read all reviews and the rebuttal comments from the authors.  I am satisfied with the authors' comments, and my score of 8 (accept, good paper) remains unchanged.

---

> > > ### Author Response · Authors · 2022-11-16
> > > **Author Reply**
> > >
> > > We thank the reviewer for taking the time to read our responses and for the recommendation.

---

### Official Review · Reviewer_H9tb · 2022-10-25

**Confidence:** 3
**Correctness:** 4
**Technical Novelty And Significance:** 3
**Empirical Novelty And Significance:** 3
**Recommendation:** 6

**Clarity, Quality, Novelty And Reproducibility:**

Originality:
I'm not familiar with the area of subset selection, but it appears that the authors have reasonably original procedures to handle the flaws of Poisson sampling (which is a standard technique with known flaws). Allowing for a differentiable subset size also seems sufficiently original.

I have some concerns that the paper ends up sounding like a series of patches to fix the flaws on Poisson sampling (e.g., lack of differentiability, high variance, etc.), but I think Poisson sampling has advantages over existing approaches, and the empirical results are convincing.

Clarity:
The paper is well written. The advantages and disadvantages of prior work and this approach are well covered, and the proposed procedure is well articulated.

Quality:
It meets the ICLR standard, and is a sufficiently good quality paper.

**Strength And Weaknesses:**

Strengths:
- The algorithm is quite elegant and simple. It also remarkably modular, as it can be used for a variety of downstream tasks, such as feature selection for explainability of downstream classifiers.

- The authors identify flaws with the Poisson sampling approach, and show algorithms that can overcome said challenges.

- I am not familiar with the state of the art for subset selection, but it appears that the proposed algorithm performs at least as well as competing approaches, is quicker, and is more flexible with respect to choice of $k$ or optimizing $k$.

Weaknesses:
- This motivation for using Poisson sampling requires some more work. There exist exact sampling methods such as weighted reservoir sampling (https://en.wikipedia.org/wiki/Reservoir_sampling#Weighted_random_sampling) that do not suffer from the problems experienced conditional  / unconditional sampling. Since the differentiable part of the Poisson network is an application of the Gumbel-Softmax trick to replace the differentiation of Bernoulli r.v.s, why not use the same technique for reservoir sampling?

- The experiments section is nice, but it seems like the three experiments are really variations of the same problem, i.e., selecting a mask that helps choose features that explain a text / image / face classifier. This surely cannot be the only application of subset selection --  what are the other standard baseline applications of subset selection?



**Summary Of The Paper:**

This paper proposes a new technique for subset sampling via conditional Poisson sampling. Given a universe of $n$ elements and inclusion probabilities $p_i$ for each element $i$, the problem is to sample a subset of size $k$ that respects the inclusion probabilities.

Poisson sampling samples each element with probability $kp_i$, leading to set of size $k$ in expectation, while conditioninal Poisson sampling would repeat unconditional Poisson until a set of size $k$ is achieved. Both of these have limitations, as uncoditinoal sampling only produces a set of size $k$ in expectation ( and may have high variance) and if $k p_i \geq 1$, element $i$ is always included; conditional sampling may not respect the inclusion probabilities.

This paper suggests techniques for handling these various issues, along with a differentiable variant that can be used in a gradient-base learning framework. Additionally, the subset size $k$ can also be made differentiable, which is useful when the exact subset size is not known and needs to be optimized, for e.g., when selecting features that are most important for downstream classification models.

**Summary Of The Review:**

My score is based primarily on the fact that the proposed approach is:
- modular
- quick
- flexible in choice of subset size $k$, and also allows for optimizing $k$ in a differentiable manner
- can be used for selecting interpretable features for downstream classification tasks

That being said, I am not an expert in subset selection and do not know much about current state of the art, and my score of weak accept is based on my current understanding of the paper. I'm happy to increase my score based on the author feedback and discussion.

---

> ### Author Response · Authors · 2022-11-12
> **Author Reply**
>
>
> We thank the reviewer for the review and suggestions.
>
> >*Weaknesses:*
> >>
> >>    *This motivation for using Poisson sampling requires some more work. There exist exact sampling methods such as [weighted reservoir sampling](https://en.wikipedia.org/wiki/Reservoir\_sampling#Weighted\_random\_sampling) that do not suffer from the problems experienced conditional / unconditional sampling. Since the differentiable part of the Poisson network is an application of the Gumbel-Softmax trick to replace the differentiation of Bernoulli r.v.s, why not use the same technique for reservoir sampling?*
> >>
>
> The main baseline algorithm that we compare against, RSS ([Xie and Ermon (2019)](https://arxiv.org/abs/1901.10517)) is in fact a reservoir sampler as elaborated in the referenced paper.
> The only difference is that for neural network applications RSS considers the finite universe and non-streaming case.
> They show that the keys generated in weighted reservoir sampling and those generated by Gumbel random variables are related by a monotone transformation which means that the order does not change.
> However, this method requires us to sort the keys, either by using a priority queue as in classical reservoir sampling or by using a differentiable Top-k operator as done by RSS.
> The use of the Top-k relaxations with large subsets, as we discuss in our paper, causes difficulties with long (relaxed) gradient chains, and time and memory.
> Furthermore the subset size parameter $k$ is not differentiable with this method.
>
> One correction we would like to make here is that we do not use Gumbel random variables with our method, but rather use straight-through gradient estimation.
> It is possible to use Gumbel-Sigmoid relaxations with this method, however, we do not take that approach here.
>
> We clarify the above in Sections 3.3 and 4.
>
> >>    *The experiments section is nice, but it seems like the three experiments are really variations of the same problem, i.e., selecting a mask that helps choose features that explain a text / image / face classifier. This surely cannot be the only application of subset selection -- what are the other standard baseline applications of subset selection?*
> >>
>
> While apparently similar in that they all relate to feature subset selection, there is a qualitative difference between the experiments in Sections 5.1-2 and 5.3.
> The experiments in 5.1-2 are on per-instance feature selection for explainability.
> The experiments in 5.3 is a subsampling experiment and selects a global set of features for downsampling and is closer to signal processing applications (Huijben et al. (2020)).
>
> However, we also show an experiment, albeit with smaller subset sizes ($k=9$), for differentiable $k$ nearest neighbour classification on MNIST and CIFAR-10 with deep network features in Appendix E.5.
> There we show that our method is on-par with the best order sampling methods.
> In another application of subset sampling Xie and Ermon (2019) also consider neighbor embeddings as an alternative to t-SNE, but again with small subset sizes ($k$=12).
>
> Nevertheless, feature selection (either for explainability or subsampling) with large instance and subset sizes shows much clearer examples of the requirement of scalability for solving the problems, which is why we focus on these problems.
>
> --
>
> We hope to have addressed the reviewer's concerns. If there are other issues which require clarification, please let us know.

---

### Official Review · Reviewer_oHmW · 2022-10-26

**Confidence:** 4
**Correctness:** 4
**Technical Novelty And Significance:** 4
**Empirical Novelty And Significance:** 3
**Recommendation:** 6

**Clarity, Quality, Novelty And Reproducibility:**

This work is overall well-written and easy to follow. It provides sufficient background for the readers to understand the proposed approach.

**Strength And Weaknesses:**

I find the most compelling part of the proposed algorithm is scalability as shown in the subsampling large image experiment where some baselines fail. The technical details seem solid to me. Still, there are a few concerns:
- My main concern is that the subset size being k is not a hard constraint. Even though Figure 1 shows that increasing the number of iterations in the conditional Poisson sampling process might mitigate this issue and result in a distribution over subset sizes being more concentrated on the desired subset size, this is at the price of high computational cost and it is not guaranteed that all the sampled subsets would have their subset size being exactly k. While this guarantee is achieved by the other existing subset sampling approaches such as RSS.
- I wonder for the conditional Poisson sampling, what is the expected number of iterations to sample a subset with a size being exactly k? Besides, while Proposition 3.1 is about the decreasing rate of variance, I wonder if this result can be generalized to the case of arbitrary input probabilities instead of the equal input probability assumption.

**Summary Of The Paper:**

This work aims to propose a differentiable and scalable k-subset sampling algorithm based on conditional Poisson sampling. The scalability comes from the conditional Poisson sampling scheme where each instance is sampled independently such that the vectorized complexity of the proposed algorithm is independent of the subset size k. The differentiability is achieved by the use of a straight-through gradient estimator. It further carries out experiments on learning to explain text and image, image subsampling, and k-nearest neighbor search.

**Summary Of The Review:**

The proposed approach demonstrates good scalability. My main concern is that the sampled subset size is not guaranteed.

---

> ### Author Response · Authors · 2022-11-12
> **Author Reply**
>
> We thank the reviewer for the review and suggestions.
>
> >*I find the most compelling part of the proposed algorithm is scalability as shown in the subsampling large image experiment where some baselines fail. The technical details seem solid to me. Still, there are a few concerns:*
> >>
> >>    *My main concern is that the subset size being k is not a hard constraint. Even though Figure 1 shows that increasing the number of iterations in the conditional Poisson sampling process might mitigate this issue and result in a distribution over subset sizes being more concentrated on the desired subset size, this is at the price of high computational cost and it is not guaranteed that all the sampled subsets would have their subset size being exactly k. While this guarantee is achieved by the other existing subset sampling approaches such as RSS.*
> >>
>
> Indeed, the method does not have a hard constraint on subset size but only on the expected size.
> This can be a limitation  when a precise subset size is needed, and it is what allows us to scale the subset sizes efficiently.
>
> With respect to variance, we emphasize that reduction variance in the subset size distribution by iteration is quite cheap.
> Due to exponential variance reduction (Proposition 3.1 and Figure 1) we only need to repeat the procedure a constant number of (no more than 8 to 10) steps to get a very small error in subset size.
> The parallel (vectorized) complexity of the steps themselves is constant, $O(1)$, up to reduction operations such as summation which require logarithmic parallel complexity (Section 3.4 in updated draft).
> This means that the overall parallel complexity is constant up to the cost of reduction operations which is quite cheap.
> In particular the complexity does not depend on the subset size as opposed to RSS.
> In Appendix E.4 we show that only a few steps are necessary to get close to the target subset size in absolute terms (i.e., minimum and maximum sizes are close to mean).
> In Appendix E.3 we show that the empirical time per epoch is independent of $k$ and the time with iteration is only slightly higher than the baseline case of a single iteration of Poisson sampling.
>
> >>    *I wonder for the conditional Poisson sampling, what is the expected number of iterations to sample a subset with a size being exactly k?*
>
> We mention the expected number of iterations for conditional Poisson sampling at the end of Appendix B.
> The number of iterations until success for conditional Poisson sampling can be seen to follow a classic geometric distribution. If $q$ is the probability obtaining a sample with size exactly $k$ in a single step, then the probability the conditional Poisson sampling procedure ends in T steps is $p(T) = (1-q)^{T-1} q$. This is a geometric distribution with expected value $1/q$.
> For the simple case of equal probabilities we can obtain $q$ using the binomial distribution.
>
> For conditional Poisson sampling, the expected number of steps until termination can be high because the geometric distribution is memoryless and the conditional Poisson sampling procedure is always restarted from scratch upon failure.
>
> On the other hand we are able to achieve exponentially decreasing variance by reusing the sample obtained in the previous step instead of the memoryless conditional Poisson method.
>
>
> >> *Besides, while Proposition 3.1 is about the decreasing rate of variance, I wonder if this result can be generalized to the case of arbitrary input probabilities instead of the equal input probability assumption.*
> >>
>
> A bound for general probabilities can be obtained by a very simple modification of the argument. We only have to replace the Binomial variance bound by the Poisson sampling variance bound in the proof (Appendix B).
>
> Under the same assumption that the normalized probabilities at each step are bounded by $q\_i$, $q\_i < 1-\epsilon$, for $\epsilon > 0$,
> we can replace the Binomial variance bound by the Poisson sampling variance bound:
> $$\sum p\_j (1-p\_j) < n q\_i.$$
> The rest of the argument remains unchanged.
>
> --
>
> We thank the reviewer for the review and hope that concerns have been addressed.

---

### Official Review · Reviewer_iTJN · 2022-11-02

**Confidence:** 3
**Correctness:** 4
**Technical Novelty And Significance:** 3
**Empirical Novelty And Significance:** Not applicable
**Recommendation:** 8

**Clarity, Quality, Novelty And Reproducibility:**

The paper is overall very clearly written and easy to follow. I only have a couple of minor comments:

1. The claim that exactly $k$ samples may not always be needed seemed a bit strong when it was first mentioned near the end of Section 1. I would recommend adding an example at that point to clarify why one could do without exactly $k$ elements.

2. Please add standard deviation/standard error values to the results in Table 1 and 2.

**Strength And Weaknesses:**

Strengths:

1. The approach is elegant, intuitive and also theoretically proved to have low (exponentially decaying) variance.
2. There are clear empirical improvements over prior work in a wide range of applications including some applications like subsampling high resolution images which are computationally infeasible with prior work (due to high memory requirements).

Weaknesses:

1. The approximate (straight-through) gradients may be too far from the true gradients. While straight-through gradients appear to be doing well in the current set of experiments, will it be feasible to explore some of the other gradient estimators mentioned, even just for the same tasks, to comment more conclusively on the right estimator?

2. The approach lacks a discussion of convergence rates. Specifically, it would be interesting to see some tail bounds on the deviation from the mean after T iterations but it is fine to leave that for future work.

**Summary Of The Paper:**

The paper proposes a new approach to sampling subsets from a set of $n$ elements using an iterative version of Poisson sampling. This avoids the high variance of standard Poisson sampling and the high cost of conditional Poisson sampling by iteratively adding or deleting elements a finite number of times. The lower variance and computational efficiency of the approach are also empirically validated across a range of experiments.

**Summary Of The Review:**

The paper introduces a novel approach for subset sampling that has low variance and is significantly more scalable than prior approaches. The approach is theoretically principled and outperforms baselines on a range of empirical experiments. It would be interesting to see some convergence rates for the deviation of the subset size from the expected size in a future work.

---

> ### Author Response · Authors · 2022-11-12
> **Author Reply**
>
> We thank the reviewer for the appreciative comments.
>
> >>   *The approximate (straight-through) gradients may be too far from the true gradients. While straight-through gradients appear to be doing well in the current set of experiments, will it be feasible to explore some of the other gradient estimators mentioned, even just for the same tasks, to comment more conclusively on the right estimator?*
>
> Using other estimators is certainly a possibility.
> For instance, there should be no real obstacle with using a Gumbel-Sigmoid relaxation for working with Bernoulli random variables.
> We touch on this in Section 3.3 and will try to explore the options in for later versions of the paper.
>
>
> >>  *The approach lacks a discussion of convergence rates. Specifically, it would be interesting to see some tail bounds on the deviation from the mean after T iterations but it is fine to leave that for future work.*
>
> Thank you for the suggestion. We do show an exponential decay in variance after $T$ steps in proposition 3.1 for a simplified case.
> Since the variance is exponentially small after $T$ steps, even a weaker relation such as Chebyshev's inequality may be used to give a strong bound on tail probability.
> By Chebyshev's inequality $Pr(|X-\mu|> t) \le \sigma^2/t^2$.
> If $t$ is constant and $\sigma^2$ is exponentially small we get a sharp tail bound.
>
> Perhaps this is similar to what the reviewer is suggesting, then we can add this explanation to the draft as a possibility? If not further clarification would be appreciated.
>
>
> >> *The claim that exactly $k$ samples may not always be needed seemed a bit strong when it was first mentioned near the end of Section 1. I would recommend adding an example at that point to clarify why one could do without exactly $k$ elements.*
>
> Thank you for the suggestion. We now point to an example of sampling large numbers image pixels for explanation in the updated draft in Section 3.1.
>
> >> *Please add standard deviation/standard error values to the results in Table 1 and 2.*
>
> We plan to add error bars to the results in the final version of the paper.
>
> --
>
> We thank the reviewer for the review. Please let us know if there are any further questions or concerns.

---

> > ### Comment · Reviewer_iTJN · 2022-11-29
> > **Re: Author Reply**
> >
> > Thank you for your response. I agree that Chebyshev's inequality is a good way to show a tail bound for the simplified case and it is similar to what I was suggesting. Please add this to the draft as well. I do not have any other comments, and since I had already recommended acceptance I will keep my score.

---

> > > ### Author Response · Authors · 2022-11-29
> > > **Author Reply**
> > >
> > > We appreciate the response and the recommendation. We plan to include the tail bound in the next version of the paper.

---

### Decision · Program_Chairs · 2023-01-20

**Decision:**

Accept: poster

**Justification For Why Not Higher Score:**

This paper achieves good results at subset sampling by relaxing the problem to a simpler problem; the overall proposed method is fairly straightforward if one is willing to relax the requirement that subsets of size K actually are size K.

**Justification For Why Not Lower Score:**

Consensus among all reviewers to accept

**Metareview: Summary, Strengths And Weaknesses:**

All the reviewers praised the paper's clear approach, theoretical motivation, and empirical validation. In that sense, the paper is a clear accept (with all reviewers arguing to accept). The paper introduces a scalable approach for selecting subsets of size K from N entities. The main weakness of this paper is that scalability comes at the expense of actually selecting subsets of size K, instead using a parallel method which selects subsets that have a random number of elements (with expectation K). While this can be a significant weakness, depending on the application, there are many settings (as demonstrated) where this is not a hard constraint.

**Note From Pc:**

if the above contains the word "oral" or "spotlight" please see: "oral" presentation means -> notable-top-5% and "spotlight" means -> notable-top-25%. As stated in our emails, we are disassociating presentation type from AC recommendations